# A political epistemology for extinction studies? On the ideas of preservation and replenishment

Lindsay Parkhowell 

Department of Languages and Literatures, University of Basel, Switzerland

## Perspective

causes and consequences; colonization; culture; extinction; indigenous

**Corresponding author:**
Lindsay Parkhowell;
Email: lindsay.parkhowell@unibas.ch

## Abstract

This perspective article takes up the challenge of articulating a political epistemology for extinction studies, centered around how both the systematic-scientific and mythopoetic traditions conceive of the idea of preservation. Political epistemology offers a solution to this for impasse because it asks the question of the social orientation or "end" of knowledge formations, thereby questioning what the larger goal of preservation might be. By focusing on the example of the thylacine, I outline one strand of what a political epistemology for contemporary justifications of preservation in the Museum might look like. Then I discuss how the mode of storytelling in extinction studies also conceives of preservation. Finally, I introduce the idea of replenishment as contrary to the preservation, focused on the cultural practices of Indigenous peoples in North East Arnhem Land, and ask whether new developments in the techno-scientific tradition will begin to turn to replenishment as well.

## Impact statement

My reflection article is novel for two reasons. First, while there has been much work done in de-colonial and Marxist epistemologies of science and modernity, extinction studies can still benefit from the development of cultural–political histories from a critical standpoint. I apply the method of political epistemology to these issues because, by analyzing the social embeddedness of all knowledge, I can also ask after the "end" or goal of the two principles I analyze: preservation and replenishment. What does preservation intend to achieve? And how does it reflect the society it emerges out of? By arguing that knowledge is not disinterested in the manner we usually conceive of as science, political epistemology helps to orient extinction studies towards the principles and goals that have led to, reproduced, or can ameliorate such extinction. Or, as Pietro D. Omodeo has put it in his book on political epistemology, "its double task is to reflect on the collective agendas behind current, disciplinary discourses on science and, secondly, to create the room for a conscious and deliberate inquiry, the political (and emancipatory) potential of which rests in the leap from implicit ideological passivity to cultural-political agency, as an autonomously chosen intellectual endeavor" (Omodeo, 2019, 2).

Second, my reflection article contributes to contemporary debates in extinction studies about the purpose and ethics of storytelling. Building on works by O'Key (2003) and van Dooren and Bird Rose (2016), I argue that the principle of replenishment requires a connection to law, cosmology, kinship and history that goes beyond the focus upon ethical entanglements in extinction storytelling. My argument here is provocative in the sense that, just like many forms of extinction storytelling, it is inspired by Australian Indigenous worldviews; unlike these other forms, however, it also involves an idealistic critique of what I call Western epitaphic storytelling. To put it more concretely, the poet-as-creator present in Australian Indigenous storytelling co-creates his or her world in a way that, I believe, offers more to the principle of replenishment than currently offered by extinction storytelling.

## The episteme of preservation

Recently, while working for the Australian Professor Katrina Schlunke on a project about the thylacine or Tasmanian Tiger, I came across a passage by the preeminent German Zoologist Heinz Moeller, which not only provides a justification for the preservation of extinct animals, but also argues for the priority of such preservation against their living presence in either a Zoo or – by implication, although Moeller does not mention this – the wild. His words struck me as indicative of a trend in modern science that privileges the access to and accumulation of what he calls the "systematic and phylogenetic information" (Moeller, 1997, 175) of an animal over and

above its living presence; and yet, the contradiction inherent to this process of observation and record is that the very evolutionary information which is of interest to scientists has been interrupted by extinction, as the animal can no longer develop in relationship to its environment.

In this short reflection, I am going to use Moeller's comments about the thylacine to outline the usefulness of a political epistemology for extinction studies. I am going to provide one example from early modern science which I believe provides the epistemological foundations for Moeller's reflections, as well as discuss how the dominant trend of "storytelling" in extinction studies also participates in the episteme of preservation. Finally, drawing upon Giambattista Vico's philosophy of history, I will propose that the mythopoetic faculty is foundational for all human societies and then turn to my second section, where I will reflect on how an episteme of replenishment could be applied to extinction studies. Moeller writes that it is preferable that the thylacine is dead in a Museum rather than alive in either a Zoo or the wild, because there is a larger "diversity and density" of information available to the museum visitor and scientist, as the specimen can be "spatially organized" alongside its evolutionary information in texts and images (Moeller, 1997, 175). Given the increasing likelihood of mass extinctions, it is worth asking what the socio-political end of Moeller's episteme of preservation for the purposes of information gathering and human consumption is. Without a conception of the "end-time" – the time in which either all information is collected or all information is extinct and therefore unavailable to itself as a growing record – the episteme of preservation remains stuck within what Moeller calls the "aura of the irretrievable" (Hauch des Unwiederbringlichen) (Moeller, 1997, 169).

A quote from Moeller's work illustrates how the aura of the irretrievable practically emerges in his Zoologist practice: "The restoration of the young thylacine in the Zoological Museum of the University of Heidelberg proved to be unavoidable in 1977; missing claws were replaced – a decommissioned creeping cat supplied the necessary 'spare parts' – and the lost tip of the tail could be added… As in other showcases of the collection, the focus is not on the object itself, but on an overriding theme" (Moeller, 1997, 171). The displaced animal carries the aura of the irretrievable in a duplicitous sense: firstly, as it is now extinct without means of renewal, and secondly as it is ossified for viewing consumption through an amalgamation of parts from other animals – complete, yes, but still irretrievably so (as the genuine "parts" are lost).

Second, Moeller's observations also raise the question of whom this knowledge is being preserved for, which is how the political meets the epistemological. I would like to demonstrate how Moeller's argument that "spatial organization" is one reason that dead specimens are preferable to live ones can be traced back to at least one epistemic foundation of early modern science. This will illustrate how a political epistemology of the museum viewer's gaze could be built, therefore also raising the political question of how or whether it might be changed. Moeller's argument for spatial organization recalls the early modern change represented by Giulio Camillo's Memory Theatre (unfinished, 1530–44), in which we see the spatial organization of human knowledge as being not only driven by accumulation but also as being pinpointed on the gaze of the Monarch for whom it is collected (see Yates, 1966, 142–143). The subject at the centre of the "knowledge of the world," while assimilated into our worldview today and even "naturalized" in terms of Museum visits and the necessary gaze of the tourist-spectator, is in fact a modern invention that relies upon the perspectival worldview of the Renaissance with the human viewer at the apex. The political is brought to bear on this episteme of viewership and the "spatial organization" of knowledge through the argument that the centralized gaze of the monarch has now become dispersed into the mass gaze of the consumer-subject, for whose gaze the knowledge is organized and therefore also prioritized. That is to say, while Museum gazing has become commonplace for us, in that we might not question Moeller's assumption that the animal is there for us to view (regardless of the provenance of its constituent parts), it has a history connected to imperial and democratic regimes of power, which further highlights the hierarchy behind Moeller's position. By positioning Moeller in such a way, we are also then able to construct whether, or how, we as viewers want to participate in such a project (the political in the epistemological).

Extinction studies, with its focus on storytelling represented by diverse works from Deborah Bird Rose, Donna Haraway and others, have a very different engagement with the episteme of preservation that decentres this consumer subject. As Bird Rose, van Dooren and Chrulew put it, extinction stories "draw us into conversation with a host of different ways of making sense of others' worlds. In large part, it is about, in Anna Tsing's (2011) terms, 'passionate immersion in the lives of nonhumans'" (Bird Rose et al., 2017, 4). However, just like the scientific worldview, extinction storytelling also involves an epistemic impasse, this time in regard to the transcription of natural life within written language. On the one hand, it is both interested in our being together with other creatures; on the other hand, however, the prevalence of mourning and loss in extinction storytelling leads to an epitaphic form of writing, even (for this author) to the question of whether all writing and transcription is, in some sense, epitaphic in the transfer it makes from the living reality to its semiotic reproduction. The importance of this observation for my argument is to question whether extinction storytelling can ever itself be involved in either an ethic or episteme of replenishment, that is, whether "passionate immersion" can ever move beyond preservation and become a mode of replenishment.

While this point about the episteme of preservation in extinction storytelling may seem forced, it is worth making because the very emergence of such storytelling happens within the socio-political context of loss. Were the natural world to be inescapably abundant and generative, extinction storytelling as we know it would not exist; furthermore, from the point of view of political epistemology, it is fair to ask whether the goals of making kin, staying with the trouble, and the passionate immersion in the lives of non-humans aim to contribute to the awareness of extinction through the modes of witnessing and inter-connectivity, rather than to isolate and change the socio-political context that has led to the increasing relevance of these modes (that is, human-driven mass extinction). For O'Key, for instance, extinction studies scholars "have offered no detailed discussion, no explicit examination, of the Sixth Extinction as a crisis of capitalism that is intensifying in the wake of the period known as the mid-twentieth century's 'great acceleration'" (see O'Key, 2003, 153–156).

Is it possible, then, to think of "preservation" in terms that do not confirm the background reality of extinction itself, but that relate to it in a way that is materially and culturally replenishing? That is to say, can a story ever become or be a mode of creation itself rather than a mode of witnessing, transcription, accumulation or information presentation in a museum? In order to answer these questions, I will now turn to what I believe to be an alternative to the episteme of preservation, that of replenishment, by discussing two texts from extinction studies in relation to Indigenous cultural practices in North East Arnhem Land, Australia.

## The episteme of replenishment

What, then, is a story? For the Italian philosopher of history Giambattista Vico, storytelling and mythmaking lay at the foundation of all human culture. He wrote that "the most sublime labour of poetry is to give sense and passion to insensate things, and it is characteristic of children to take inanimate things in their hands and talk to them in play as if they were living persons … This philological-philosophical axiom proves to us that in the world's childhood men were by nature sublime poets" (Vico, 2002, XXXVII). Vico's description of the mythical past as "childish" and therefore of the written present as "mature" is misleading, for his philosophy in The New Science is only superficially a developmental one; rather, he means to show how the human myth-making faculty is foundational and present even in rationalist societies.

This becomes relevant once the first question I asked earlier, of what objects in Museums are being preserved for, is raised. If human societies are essentially myth-making, then preserving objects to assemble and present their "systematic and phylogenetic" information, as Moeller argues, is only a second-order aim that is subordinate to a larger mythical project. I would suggest that the collection and preservation imperatives of colonialism and neo-colonialism respectively both follow the same goal, which is the accumulation of objects into centres of power to increase their cultural capital. I suggest that this is why the scientific preservation of information is not equally interested in the preservation of the lifeworld of the animal, as the latter can only be transformed into capital through means like tourism rather than re-wilding. I would direct those readers who think this step is too elaborate to recall Moeller's "aura of the irretrievable", which he writes haunts attempts to portray or understand the thylacine. This pseudo-mythical project of preservation without recovery evokes a scientific version of millenarianism, in which what has been forever lost haunts the continuing project of its preservation.

Privileging myth-making, as Vico asks us to do, turns us back to more explicit mythmaking cultures in order to find a contrary or supplement to the idea of preservation. For Indigenous peoples living in East Arnhem Land, Australia, rock and cave art are not seen as "distant" historical objects that represent evolutionary information, but rather as living artifacts of a connection to the country. I raise this example because it is essentially non-auratic and therefore at odds with post-Enlightenment Western conceptions of art; the keepers or guardians of the mythical tradition that is represented have the duty of "updating" or "touching up" the representations of this tradition, as they are a reciprocal part of its living record. While, to the best of my knowledge, this does not relate to creation myths that are no longer actively practiced in the community – for example, the art from the Pilocene period – it does teach us an important lesson about the connection to myths as being actively replenished by the human creator-poet rather than fixed in evolutionary time. Because my first reviewer has pressed me to explain this point further, I ask the reader to imagine what it would be like if the elders of Florence gathered together yearly to spiritually renew and even change the unfinished sculptures of Michelangelo – an impossible thought.

This example from Australia sheds light on the several questions I raised earlier about the episteme of preservation. First, as the human being is no longer centered as sovereign over nature, but rather as a part of its circular renewal, replenishment is a mode of relation rather than simply the content of adjusting or improving artworks or objects. This circular renewal is eternal with the land, or country, as the first ancestor that then gave rise to myths, which in turn structure the complex kin relations and laws of Indigenous communities. One example might be that if a certain skin group has suffered a loss recently, (say, in the flying fox population), then they will not hunt them for a while so the population can recover. In art, Josie Maralngurra, the granddaughter of famous artist Nayom-bolmi, has described the creation stories of rock art as a "vehicle of memory" that crosses generations and is essentially participatory instead of the spectatorship of the museum (Rock Art, 2022). This mode of relation is not epitaphic because it belongs to an essentially oral culture, which not only means that traditions and knowledge are passed down through kin relations, but also that those people then become themselves the "keepers" of the stories or objects, not their witnesses or transcribers. While extinction storytelling also draws on such traditions and ethical modes of relation, it does not go so far as to "belong" to that animal group, however, involved its relations may be. For this to be true in the Western world, we would have to imagine artworks that produce an instant moratorium on mining practices that destroy the habitats of birds (for instance).

I will give two examples from extinction studies in order to deepen my argument. The first comes from Thom Van Dooren and Deborah Bird Rose's article Lively Ethnography: Storying Animist Worlds from 2016. In this article, the authors advance two principles of lively ethnography: a style of writing that is both expository and performative in bearing witness to more-than-human world(s), and an ecological animism that "responds to a world in which all life – from the smallest cell to the largest redwood – is involved in diverse forms of adaptive, generative responsiveness" (Bird Rose and van Dooren, 2016, 82). The challenge that arises from this sophisticated and inspiring attempt to tell extinction stories in a restorative and interconnected way is that, unlike the Australian Indigenous animism it draws upon, there is no broader societal connection to law, cosmology, history and kinship in the stories that are told. This argument is not forced out of the two different cultural circumstances; to "make kin" is to also be co-produced as a subject in a world in which that kinship is not a choice, ethical approach, or philosophical position, but involves irresistible obligations (law) as well as cosmology and history (the stories as creation myths and historical record); it is a fundamentally different worldview in which the stories are themselves the mode of creation and relation instead of a record of them, however deeply felt and researched. This shift is so intricate that it bears repeating even if I cannot fully outline – as Bird Rose and van Dooren have so helpfully done – how it would develop in a Western-based society.

The second example comes from a very different sort of text, Deborah Bird Rose's Wild Dog Dreaming: Love and Extinction from 2011. Having spent a long time with Indigenous Australians, Bird Rose is in a unique position to reflect upon the intersections of Indigenous and Western worldviews and philosophies. While much of Wild Dog Dreaming: Love and Extinction is concerned with the Dingo as a sacred creator-spirit and animal-kin in Indigenous life, it also discusses the killings of Dingoes as either a "pest" animal who threatens pastoralism, or the outright murder of them in pastoralist assaults. For Old Tim, her Indigenous mentor, it is very clear that "Dog's a big boss, you've gotta leave him be, no more killing" (Bird Rose, 2011, 65). My proposal to advance the episteme of replenishment over that of preservation has been motivated by hearing similarly profound statements that can so humbly observe the limit on human behavior even in the face of great crisis or pain.

Chapter 7 of Wild Dog Dreaming: Love and Extinction features a comparative discussion of the Book of Job and the Dingo creator-

myth, and comes the closest to an episteme of replenishment in extinction storytelling that I have found so far. Bird Rose writes of the myth of the Moon, who is able to replenish himself every day after dying and cajoles the Dingo into trying to follow him into death. The Dingo eventually relents but cannot come back, as the Moon can; and yet, he is called for and missed by his friends, which the Moon, in his solitude, doesn't share. For Bird Rose, this story is an example of how "animals and plants, all our precious Earth mates, are abandoned as they tumble into deaths that have no return… Who sings out to them?…along with the activism required of us in these days of grief, let us not forget to keep singing" (Bird Rose, 2011, 79-80). Because of the Dingo's role as the "first" to die in this myth, the story also serves as a model for the rest of humanity, and for grieving practices in which the dead are called until their spirits have gone where we cannot follow.

For my argument, however, the story highlights how the "myth" of replenishment in the face of death – the calling and singing to spirits who are gone, who then also become transformed into other animals after dying – is still far away from the foundational myths of Western modernity, which we have to grapple with if we are to contest them properly. What would it mean, for example, to tell ourselves a story about some of the great political myths of modernity, such as the "myth" of alienation or the "myth" of progress, in such a way that we would become both the creator-poets of such stories with the agency to change them, and the humble listeners of them, the observers of the limit, or the law, in the prevention of violence to others? Indigenous Australian storytelling is replete with such stories, discussed by Bird Rose in this book and many others. The challenge that I am trying to advance, which I think Bird Rose is also grappling with by re-telling this Dingo creator-myth, is how it is possible for such stories to influence the wider laws of a society, and not retrospectively – after the loss has happened – but foundationally, in order to prevent them in the first place. I believe this is why Bird Rose ends her chapter with a song to call the extinct animals back, which could become – in its living presence, if this song was sung by many people, for example in the opening to a Parliament or political movement – more than an epithapic record or witnessing of loss.

Finally – and this is merely speculation – the episteme of replenishment may well also enter the discourse of museum studies, as the biotechnology company Colossal has announced that they have developed technology to bring back extinct mammals, including the mammoth, Tasmanian tiger and the dodo. What would a park of (living) extinct animals look like, and how would it influence the viewers who visit it? Would the gaze of the consumer subject be valorized over and above the lifeworld of the animals themselves, or would it create a haunted space that could lead to mourning those losses of extinction brought about by colonialism? I end with this speculation since I believe it provides a contribution to the growing discourse of repair in decolonial and museum studies.

## Conclusion

As I have written, political epistemology inquiries into the "end" or goal of knowledge formations by assuming the social embeddedness of all knowledge (see Omodeo, 2019). I have analysed one possible epistemological foundation for the preeminent German zoologist Heinz Moeller's remarks that it is preferable that a thylacine be kept in a collection rather than a Zoo or in the wild: its emphasis on the gaze of the consumer-subject (or scientist) in the Museum, who has inherited the centralizing gaze of the Monarch present in Giulio Camillo's Memory Theatre. Furthermore, I have analysed how extinction storytelling also relies upon an idea of preservation, albeit one that refers now to an epitaphic witnessing from the lifeworld of an animal to its transcription. For extinction studies to be truly replenishing in the Indignous sense – and here I am speaking ideally – it would need to understand the "transcriber" as themselves a creator-poet so connected or related to the animals or myths that they also have the right, and the ability, to change it. Their works would need to be culturally relevant enough so that they could return to their "tribe" (Western law) and change the circumstances leading to extinction.

This research into translating documents about thylacine collections in Berlin was supported by the ARC Discovery Grant DP2000101877 'After Extinction: Reconstructing the Global Thylacine (Tasmanian Tiger) Archive' led by Dr. Katrina Schlunke.

**Open peer review.** To view the open peer review materials for this article, please visit http://doi.org/10.1017/ext.2025.3.

**Acknowledgements.** The author is the sole contributor to this article, and would like to thank their first reviewer for the excellent feedback.

**Competing interests.** The author declares none

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
