## [Reviewer Report]

Page 1

What does the author mean by “political epistemology”? They do not define it in their impact statement.

The author states that the scientific understanding of preservation is “either taxidermic or scholarly” (24/25). Does this obscure the ways that preservation also sometimes functions synonymously with conservation?

The author asserts a stark and generalising dichotomy between Western and Indigenous epistemologies (31) which is not at this point substantiated enough. I’m really struck, though, by this articulation of preservation vs replenishment, and I look forward to reading more about it across the paper.

The author motivates the novelty of their paper by claiming that extinction studies (which they do not define in their impact statement, but instead gesture at is as a “broad area” of inquiry) has not interrogated “epistemologies of science and modernity” (36). Yet this is not evidenced sufficiently, as both of the extinction studies texts they will later reference make substantive critiques of science and modernity.

The author says that their article will help orient extinction studies “away from just a study of extinction” (45). But why is a study limiting? What’s at stake in moving beyond it? Why should extinction studies be more than a study? And what evidence –beyond its name– does the author have for arguing that these works are just studies?

Page 2

(7) “As well” ; should be “as well as”

Why does the author italicize episteme?

(15) Moeller’s theory is as fascinating as it is troubling and self-defeating. So much so that I think it requires much more unpacking and deliberation, especially concerning his rather gnomic formulation “the aura of the irretrievable”, which the author does not gloss for the reader. Could the author consider introducing Moeller in more detail, and giving a longer, block quote from Moeller that gives the reader a sharper sense of his peculiar perspective?

(32) The sentence “Bacon’s ideas, also represented by his famous phrase…” is too long, packing in many ideas that require explication.

Why does the author turn to two early modern thinkers, Francis Bacon and Guilio Camillo, to help us understand the reflections of a twentieth century zoologist and subsequently the field of extinction studies? More explanation and justification is needed.

(39) The paragraph on Camillo is rather cursory, which leaves the reader feeling like the connection and precedent is tenuous rather than crucial.

(53/54) Bird Rose does not have a hyphen or dash; Haraway misspelled; repetition of “represented”/“represents”

Page 3

Does extinction studies “lead … even to the question of whether all writing…” (5), or does it lead you to this question? It’s unclear.

(12) “Were the natural world to be inescapably abundant and generative, extinction storytelling as we know it would not exist as we know it”. While I do not want to pedantically dispute this, as it’s certainly right, I do wish to say that because there’s an ever-present background rate of extinction, even in biodiversity rich times, it’s not inconceivable that we might still have a form of extinction storytelling, albeit a very different one

The author states that “it is fair to ask” (14/15) whether extinction studies is more a project of preservation than political change. This is a similar point to one I have made in an article for Environmental Humanities (O’Key, “Extinction in Public” 2023).

(22) “that is to say” – capitalise That.

(24) “believe to an” – believe to be?

(31) “He wrote that” does not require the subsequent comma

(47) why are “objects” parenthetically redescribed as “capital”? Capital is its own specific thing, process, and system.

Page 4

(3) “at odds with any Western conception of art” – the multifaceted histories of experimental, avant garde, subcultural, queer and working class artistic movements would beg to differ.

(11) again, episteme is italicised.

(12) idea of circular renewal is crucial and needs explaining more deeply.

(17) “In contradistinction to extinction storytelling …” I’m really struck by the author’s claim here, which seems to miss the fact that Bird Rose’s Wild Dog Dreaming is deeply engaged with indigenous cultures and notions of country.

(22–28) This paragraph is quite weak, referencing an anonymous and homogenous “they” and offering self-proclaimed “impressions” about resilience that stray towards patronising romanticism.

(30–35) The author gives an example that is on display in the National Gallery of Victoria. Does the artwork’s institutionalisation and consecration have any bearing on the argument they are making, considering the author’s earlier claims that the museum is a spatial organisation of preservation?

(40) “Thompson nevertheless…” how does he show this?

(43–50) I would suggest cutting this paragraph on technology as it does not advance the argument and comes out of nowhere.

Page 5

(6) The author again says that extinction storytelling is “epitaphic”, but here – as in above – they do not give any evidence of this from extinction studies itself.

My concluding thoughts:

This position paper has promise. Its theoretical conceptualisation of replenishment as a decolonial counter to modernity’s preservationism is very interesting and powerful; its critique of extinction studies as being part of the episteme of the latter, rather than the former, is similarly provocative. Yet as things stand these two points are merely made rather than convincingly argued.

For this paper to be developed, the author could consider developing their critique of extinction studies by giving evidence from its publications, citing more widely from the field and even incorporating works that already pose questions about the field (e.g. Donahue, “Survival and Extinction”, 2021; O’Key, “Extinction in Public”, 2023). They might also spend more time developing the relationship between Moeller’s zoological empiricism and poetic reflections on the one side and extinction studies on the other. More crucially, the author should revisit their dichotomizing assertions about a “contrast” between western and indigenous epistemes, either to bolster this argument with more evidence or else complicate its generalizing binary logic. Finally, this paper would be improved if the author focused their attention on the paper’s strengths, namely: theorising political epistemology and replenishment.

---

## [Editor Report]

First of all, I would like to apologise for the long delay in reviewing your paper. We were ultimately unable to allocate a second reviewer, so we have assessed your paper on the basis of a single, quite comprehensive reviewer’s report. I agree with the reviewer that the central critique of the paper is a powerful one, and that the paper has the potential to make a strong contribution to the Special Issue pending the major revisions detailed in the review. Alongside the reviewer, I would like to see a more rigorous characterisation of “political epistemology” and integration of recent work in Extinction Studies that deals explicitly with related epistemological issues around public display, particularly O’Key’s Extinction in Public. The paragraph 22-28 on page 4 is problematic with its reference to an unspecified Indigenous “they” (and Indigenous should be consistently capitalised throughout). We look forward to reading a revised version of your paper soon.

---

## [Reviewer Report]

1) The author still asserts a generalizing dichotomy between Western and Indigenous ideas. It’s clear from their response letter that they’re quite committed to this dichotomy. What would be helpful, then, is a gloss on what the author means by indigeneity. Put simply, briefly theorize indigeneity. The author focuses on the Australian context, but they don’t explain whether they mean indigenous as a global category, or as one (productively) limited to the case study they offer. The link between the Australian context and a concept of global indigeneity remains presumed, rather than explained. Relatedly, I’m still unpersuaded by the totalising assertion that every “Western concept of art” is auratic. You could perhaps qualify this by identifying auratic art as the dominant or major force of post-enlightenment aesthetics, for example. Or give us evidence as to why you’re right, rather than positing the claim as accepted truth.

2) The essay claims that extinction studies reproduces the preservationist paradigm. It concludes with a fascinating and powerful point that “For extinction studies to be truly replenishing...". To what extent is this already achieved by Bird Rose’s book Wild Dog Dreaming, and by articles like this one (https://read.dukeupress.edu/environmental-humanities/article/8/1/77/61701/Lively-EthographyStorying-Animist-Worlds)? If not, tell us how and why, with citations to these texts that show how they remain transcriptions and witness statements.

3) The two examples provided on page 5 are rather descriptively abrupt and risk an unacknowledged analytical reduction. The literary text, for example, is reduced to a claim about one short story’s plot. I’m sympathetic towards the need for brevity, however the essay would benefit from giving more space to these case studies so that they don’t appear, as they do now, as mere confirmations of the argument.

4) local observations:

* p.1 51 “auton-omously” ?

* p.2 12 provide, 58 passive voice without a subject: “by arguing”,

* p.3 Bird Rose … Bird-Rose

* p.4 40 I assume it should be kin, not skin?

* p.5 “influece”

* Across the essay more generally, the author ought to look over the formatting of their punctuation when using quotations. I suspect there are come copy-pastes here in which quotation marks are rendered straight, and end-of-line word breaks are reproduced.

* They also have inconsistent capitalisation and italicisation practices, e.g. museum/Museum, and thylacine italicised.

---

## [Editor Report]

Thank you for your submission of a revised version of this essay, which reads well. The reviewer has recommended a set of minor revisions to be made before the essay can be published- I agree that making the suggested revisions (1-4) will add nuance and consolidate your arguments, especially around Indigenous worldviews. This should include fixing the editorial issues throughout the essay, including inconsistencies in capitalisation throughout (e.g. Indigenous) and errors in spelling and punctuation (e.g. Arnhem Land at the top of p. 5, Bird Rose, etc). You can ignore the reviewer’s comment about skin-groups on p. 4. I look forward to seeing the final version!

---

## [Editor Report]

I’ve read this resubmission in light of reviewer 1’s comments. There is still a suggestion in the essay that all Western art is non-auratic, but there has been a serious attempt to theorise Indigeneity in the context of Bird Rose’s work and I find the revised essay generally interesting and persuasive. (Note that Indigenous is misspelled in the final paragraph). I’m happy to recommend publication. Thank you for your patience during the editorial process.